# Assessment of Biofilm Formation and Anti-Inflammatory Response of a Probiotic Blend in a Cultured Canine Cell Model

**DOI:** 10.3390/microorganisms12112284

**Published:** 2024-11-11

**Authors:** Nicholas L. F. Gallina, Nicole Irizarry Tardi, Xilin Li, Alvin Cai, Mandy J. Horn, Bruce M. Applegate, Lavanya Reddivari, Arun K. Bhunia

**Affiliations:** 1Molecular Food Microbiology Laboratory, Department of Food Science, Purdue University, West Lafayette, IN 47907, USA; ngallin@purdue.edu (N.L.F.G.); irizarrn@purdue.edu (N.I.T.); li2807@purdue.edu (X.L.); cai325@purdue.edu (A.C.); 2Purdue Institute of Inflammation, Immunology, and Infectious Disease, Purdue University, West Lafayette, IN 47907, USA; applegate@purdue.edu (B.M.A.); lreddiva@purdue.edu (L.R.); 3CH2 Animal Solutions, 21 Bear Creek Estates Dr., Ottumwa, IA 52501, USA; mandy.horn@ch2animalsolutions.com; 4Purdue University Interdisciplinary Life Sciences Program, Purdue University, West Lafayette, IN 47907, USA; 5Department of Food Science, Purdue University, West Lafayette, IN 47907, USA; 6Department of Comparative Pathology, Purdue University, West Lafayette, IN 47907, USA

**Keywords:** probiotics, biofilm, antimicrobial activity, adhesion, pathogens, canine gut health, anti-inflammatory response, MDCK, cultured cell model, *Lactobacillus*, *Enterococcus*

## Abstract

Gut dysbiosis and an inflamed bowel are growing concerns in mammals, including dogs. Probiotic supplements have been used to restore the natural microbial community and improve gastrointestinal health. Biofilm formation, antimicrobial activities, and immunological responses of probiotics are crucial to improving gut health. Thus, we tested a commercial probiotic blend (LabMAX-3), a canine kibble additive comprising *Lactobacillus acidophilus*, *Lacticaseibacillus casei*, and *Enterococcus faecium* for their ability to inactivate common enteric pathogens; their ability to form biofilms; epithelial cell adhesion; and their anti-inflammatory response in the Madin-Darby Canine Kidney (MDCK) cell line. Probiotic LabMAX-3 blend or individual isolates showed a strong inhibitory effect against *Salmonella enterica*, *Listeria monocytogenes*, enterotoxigenic *Escherichia coli*, and *Campylobacter jejuni*. LabMAX-3 formed biofilms comparable to *Staphylococcus aureus*. LabMAX-3 adhesion to the MDCK cell line (with or without lipopolysaccharide (LPS) pretreatment) showed comparable adhesion and biofilm formation (*p* < 0.05) to *L. casei* ATCC 334 used as a control. LabMAX-3 had no cytotoxic effects on the MDCK cell line during 1 h exposure. The interleukin-10 (IL-10) and tumor necrosis factor alpha (TNFα) ratio of LabMAX-3, compared to the *L. casei* control, showed a significant increase (*p* < 0.05), indicating a more pronounced anti-inflammatory response. The data show that LabMAX-3, a canine kibble supplement, can improve gastrointestinal health.

## 1. Introduction

Probiotic application is widespread in humans and animals for their anti-inflammatory and immunomodulatory response, as well as improved gut health [1,2,3]. The definition of probiotic in the root sense means “for life”, and the critical expectation is that live and active cultures should be delivered in substantial amounts to observe health benefits [4,5]. Some probiotic bacteria are predominant residents of the gut [6]. Probiotics have desirable attributes that benefit the host, such as producing short-chain fatty acids (SCFAs), immunomodulation, surviving acidic environments, pathogen exclusion, and influencing the microbiome composition [7,8]. Probiotic bacteria have been used in human food or animal diets to prevent pathogen colonization and improve gut health through competitive exclusion [9,10,11]. Probiotics are now used as an additive to kibble and wet foods for companion animal gut health [2,12]. Their beneficial effects are either transient or ineffective [2,13], primarily due to their inability to colonize the inflamed bowel [14,15,16]. Probiotics have also been used to sequester mycotoxin from feed [17,18,19]. Therefore, precedence exists for exploring further characterization of probiotic additives in companion animal feed.

Canines are exposed to numerous pathogenic and environmental insults that make them vulnerable to illness. Dog food has been linked to multiple outbreaks associated with *Salmonella enterica*, *Listeria monocytogenes*, *Escherichia coli*, and other pathogens [20,21,22]. In the dysbiotic gut, pockets of inflammation and dysregulated epithelial cell tight junction permeability are evident [23]. The inflamed state of the bowel propels the reconstruction of the enteric flora to that of a negative state, termed dysbiosis [24]. Canine gut dysbiosis has shown an imbalance in the standard phyla of bacteria, including *Firmicutes*, *Fusobacteria*, *Bacteroidetes*, *Proteobacteria*, and *Actinobacteria* [25,26].

Using probiotic bacteria with their beneficial properties as a canine food additive is one of the attractive methods of reducing pathogen exposure, as well as antibiotic use and promoting gut health [2,16]. Clinical trials for probiotics in pet food have been carried out, and several products on the market contain stable probiotics that may promote pathogen clearance and improve gut health while promoting overall animal fitness [27].

Natural probiotics may have limited capabilities of colonizing the inflamed-damaged tissue; therefore, recombinant bioengineering has been used as a novel strategy to enhance probiotic epithelial adhesion, biofilm formation, immunomodulation, and pathogen colonization resistance [14,28,29]. Nevertheless, genetically engineered and native probiotics possess the same inherent properties for pathogen exclusion, such as nutrient partitioning, biofilm formation, and clusters of abundance within the gastrointestinal (GI) tract [30].

Biofilm formation by pathogenic or beneficial bacteria aids in their adhesion and colonization to biotic and abiotic surfaces [31,32,33]. Since probiotics are natural gut inhabitants, they can form a biofilm on the mucosal surface, often in mixed communities, which is a vital and desirable attribute [32]. During biofilm formation, bacteria communicate via the messaging system termed quorum sensing [32,34,35]. Probiotic biofilms promote gastrointestinal healing, community structure, and the immunomodulatory response [36]. Bacteria produce exopolysaccharides, proteins, and extracellular DNA to aid in biofilm formation [33]. Biofilms produced by probiotics can competitively inhibit pathogen colonization, such as seen with the *Listeria monocytogenes* challenge in a mouse model [14]. Furthermore, biosurfactants from *Lactobacillus jensenii* and *Lacticaseibacillus rhamnosus* have shown antimicrobial and antibiofilm formation properties against clinically relevant strains of *Acinetobacter baumannii*, *Escherichia coli,* and *Staphylococcus aureus* (MRSA) [37]. A cocktail of *Lactobacillus fermentum* and *Lactiplantibacillus plantarum* could also reduce *S. aureus* (MRSA) biofilm production [38].

Therefore, in this study, we aimed to evaluate antimicrobial activities, epithelial cell adhesion, biofilm formation, and the anti-inflammatory response of a canine probiotic blend (LabMAX-3) that contains *Lactobacillus acidophilus*, *Lacticaseibacillus casei*, and *Enterococcus faecium*, which are known to promote canine gut health [39,40,41], using a Madin-Darby Canine Kidney (MDCK) cell line. MDCK cells are derived from the renal tubule of the canine kidney and display enterocyte-like absorbing properties; thus, they are widely used as a canine gut epithelial cell model [42,43].

## 2. Materials and Methods

### 2.1. Probiotic Preparation

A commercial proprietary probiotic blend, LabMAX-3, containing equal proportions of three probiotic cultures of *Enterococcus faecium*, *Lactobacillus acidophilus*, and *Lacticaseibacillus casei*, was received from CH2 Animal Solutions (Ottumwa, IA, USA). Since the lyophilized powder blend (LabMAX-3) could not be directly used in our proposed experiments due to the presence of some insoluble materials (part of formulations), we isolated each culture from the blend and mixed them at a 1:1:1 ratio, designated LabMAX-3. The lyophilized powder was plated on DeMann Rogosa Sharpe (MRS, Thermo Fisher Scientific, Walthan, MA, USA) agar plates and incubated at 37 °C for 48 h under anaerobic conditions using a Gaspak (Thermo Fisher Scientific). The colonies were picked based on phenotype, and each bacterial culture morphology was verified by microscopy (Leica, Deerfield, IL, USA). *L. acidophilus* NRRL 31910, *Pediococcus acidilactici* H, and *E. faecium* ATCC 8459, as controls, were propagated in MRS under anaerobic conditions. *Lacticaseibacillus casei* ATCC 334 was cultivated in modified MRS agar (MMRS) containing 1% *w*/*v* proteose peptone, 0.5% *w*/*v* yeast extract, 0.1% *v*/*v* Tween 80, 0.2% *w*/*v* meat extract, 37 mM C_2_H_3_NaO_2_, 8.8 mM C_6_H_14_N_2_O_7_ dissolved in 0.2 M potassium phosphate (dibasic, pH 7.0), 0.8 mM MgSO_4_, 0.24 mM MnSO_4_, and 1% *w*/*v* mannitol and supplemented with erythromycin (2 µg/mL, Thermo Fisher Scientific) and grown in anaerobic conditions [14].

### 2.2. PCR Confirmation of Probiotic Strains

Probiotic culture isolates in LabMAX-3 were confirmed by PCR amplification of the species-specific target genes listed in Table 1. Target gene amplification was performed using PCR Master Mix (Platinum™ Hot Start, Thermo Fisher Cat# 130000). An equal volume of template (2 µL of resuspended colony) was added, and DNA was amplified via a thermocycler (Applied Biosystems, Carlsbad, CA, USA). Thermocycling conditions are as follows: 94 °C for 10 min, 30 cycles at 94 °C at 30 s per cycle, 30 s at 60 °C, 72 °C at 30 s, and 5 min at 72 °C. PCR products were analyzed by agarose gel electrophoresis (2%), and DNA bands were visualized under UV exposure using a Kodak EDAS 290 camera (Rochester, NY, USA) with ID LE 3.6.3 software.

### 2.3. Pathogen Propagation

Cultures of enterotoxigenic *Escherichia. coli* (ETEC) strains F4 and O78:H11, *Staphylococcus aureus* ATCC 25923 *Salmonella enterica* serovar Typhimurium ST1, and *Listeria monocytogenes* F4244 from our culture collection (Table 2) were cultivated in Tryptic Soy Broth supplemented with 0.6% yeast extract (TSBYE) at 37 °C for 18–24 h. *Campylobacter jejuni* ATCC 29428 was cultivated in microaerophilic conditions (5% CO_2_, 37 °C, 7 days) in Bolton Broth (Neogen, Lansing, MI, USA).

### 2.4. Antimicrobial Activity Testing on Agar Plates

The antimicrobial activity of individual probiotic cultures of *Enterococcus faecium*, *Lactobacillus acidophilus*, and *Lacticaseibacillus casei* in a mixture (1:1:1) of the three designates LabMAX-3, *P. acidilactici* H, and *L. casei* ATCC334 was tested against pathogens (Table 2) as before [46]. Briefly, overnight-grown (37 °C for 18 h) cultures of probiotic strains were inoculated (1 µL, stab method) on a base layer of MRS agar and incubated for 18–20 h at 37 °C anaerobically. Pathogens were grown at 37 °C for 24 h in TSBYE aerobically. Tryptic Soy Agar supplemented with 0.6% yeast extract (TSAYE, Thermo Fisher Scientific) soft top agar (0.8% agar) was prepared, and 10 µL of the respective pathogens were inoculated and vortexed. The inoculated soft agar (5 mL) was then poured onto the base layer, swirled, air-dried, and incubated at 37 °C for 24 h. The zones of inhibition around the stab culture were measured.

For the preparation of cell-free supernatants, probiotic bacterial culture (18–20 h grown as above) supernatants were collected after centrifugation (14,000× *g*, 5 min) and passed through 0.45 µm membrane filters. A 20 µL aliquot was tested on a sterile blotting paper disc against the test pathogens that were overlaid on TSA agar plates [46].

In separate experiments, each 18–20 h-grown probiotic culture was heat-treated (80 °C for 10 min), and 10 µL of each culture was placed on the BHI agar surface and overlaid with the test pathogens as above [46].

### 2.5. Biofilm Formation by Probiotics

Biofilm formation by probiotic blend LabMAX-3 was analyzed in multi-well polystyrene plates as before [47]. Briefly, anaerobically grown probiotic bacteria were suspended in a 1:1 ratio of MRS and MMRS, dispensed into 96-well tissue culture plates at 1.0 × 10^8^ CFU/well (TPP, Trasadingen, Switzerland), and incubated at 37 °C for 24, 48, and 72 h anaerobically. *Lacticaseibacillus casei* ATCC 334 and *Staphylococcus aureus* were used as positive controls and grown in MRS and TSBYE, respectively, and inoculated at 1 × 10^8^ CFU/well in a 1:1 ratio of MRS:MMRS or MRS:TSB, respectively, then incubated at 37 °C for 24, 48, and 72 h aerobically. Every 24 h, the old media were aseptically replaced with fresh media. Biofilms were washed three times in 0.2 mM phosphate-buffered saline, pH 7.2 (PBS), to remove planktonic bacteria. Bacterial counts in the biofilm were enumerated after scraping the biofilms from the well; collecting the cells in an Eppendorf tube; sonication for 20 min using iSonic Model #P4830 set at Frequency 60 Hz, Watt 150, Volt 110–120, and waveform 18.3 (Chicago, IL, USA); and serially diluted for plating on MRS or TSAYE agar plates [31].

The formation of biofilms was also assayed by crystal violet staining. Briefly, washed biofilms were stained with 1% crystal violet (Sigma, St Louis, MO, USA), washed three times in PBS, air-dried for 15 min at room temperature (~25 °C, RT), treated with 95% ethanol, and the absorbance (OD590_nm_) of supernatant was measured [47].

In a separate experiment, bacterial biofilms were allowed to form in multiwall chambered glass slides (LabTek II, Cat# 154534, Thermo Fisher) that were UV-pretreated for 45 min. Wells were washed 3× in PBS, air-dried for 15 min at RT, heat-fixed, subjected to Gram staining using crystal violet without the counter staining, and visualized under a Leica Microscope (Deerfield, IL, USA) with 1000× magnification.

### 2.6. MDCK Cell Line Preparation

The Madin-Darby Canine Kidney (MDCK, NBL-2, CCL-34, ATCC, Manasas, VA, USA) cell line was cultivated in Dulbecco’s Modified Eagles Medium (DMEM, Gibco, Grand Island, NY, USA) supplemented with 10% Fetal Bovine Serum (FBS, Gibco), termed D10F. The canine cell line was grown at 37 °C at 5% CO_2_, passaged from 4–15, and propagated in T-75 flasks (TPP, Switzerland) until 75% confluence was reached. The cells were treated with Trypsin-EDTA (0.25%) (Gibco), centrifuged (800× *g* for 2.5 min), and resuspended in D10F. The MDCK cells were counted by 0.4% Trypan blue (Gibco) staining and seeded at a density of 10^4^/well.

### 2.7. Treatment of MDCK with Lipopolysaccharide

Lipopolysaccharide (LPS, 1 mg/mL, Sigma, St Louis, MO, USA) was reconstituted in sterile deionized (d) H_2_O and aliquoted for one-time usage. MDCK cells were treated with LPS at 1 µg/mL in D10F for 24 h at 37 °C and 5% CO_2_ to induce modest inflammation [43]. The monolayers were washed three times in DMEM to remove the LPS present and then treated with probiotics. Control wells (no LPS) received equivalent amounts of dH_2_O to serve as a vehicle control.

### 2.8. Probiotic Adhesion to MDCK Cell Line by Plate Counting

MDCK cells were seeded at 1 × 10^4^ cells/mL/well and cultured for 9 days to allow monolayer formation. Cell monolayers were washed three times in DMEM and treated without (no LPS groups) or with freshly prepared D10F containing 1 µg/mL LPS and incubated for 24 h (37 °C, 5% CO_2_). Monolayers were then washed three times in DMEM to remove residual LPS, and cells were examined for monolayer integrity by microscope.

Probiotics were grown anaerobically for 24 h in MRS or MMRS as before, added to MDCK cell monolayers with a multiplicity of exposure (MOE) of 1000, and incubated at 37 °C with 5% CO_2_ under humidified conditions for 24 h. Supernatants were collected for lactate dehydrogenase (LDH) and cytokine profiles (see below). MDCK cell monolayers were washed three times in DMEM to remove unattached bacteria. Cell monolayers were detached using Triton-X 100 (Sigma) treatment (0.1% for 5 min), vortexed, serially diluted, plated, and incubated at 37 °C for 48 h anaerobically. Colony counts were plotted to determine bacterial adhesion.

### 2.9. Adhesion and Biofilm Analysis by Giemsa Staining

MDCK cells were seeded (1× 10^4^ cells/mL/chambered well) in UV-pretreated (45 min) cassettes (LabTek II, Cat# 154534, Thermo Fisher), incubated at 37 °C at 5% CO_2_ for 9 days, and inoculated with bacteria at MOI 1000, as above. The cell monolayers were fixed with 100% methanol for 10 min, air-dried, and stained with Giemsa stain (1:20 dilution in dH_2_O and methanol, stained for 45 min, air-dried, and visualized under the Leica microscope, Leica, Wetzlar, Germany).

### 2.10. Lactate Dehydrogenase Assay

Lactate dehydrogenase (LDH) release was from MDCK cells, and percent cytotoxicity was calculated as per the manufacturer’s instructions (Cayman Chemicals, Ann Arbor, MI, USA) [14].

### 2.11. Cytokine ELISA

Canine cytokines IL-10, TGFβ, and TNFα were purchased from R&D Systems (Minneapolis, MN, USA), and cytokine levels in MDCK culture supernatants were quantified following the manufacturer’s instruction.

### 2.12. Data Analysis

Data were analyzed with GraphPad Prism (La Jolla, CA, USA) with unpaired Mann–Whitney tests. All data sets are representative of at least three experimental/biological replicates. Data are presented with standard error of the mean (±) or box and whisker plots with an interquartile range.

## 3. Results and Discussion

### 3.1. Antimicrobial Activity of Probiotics Against Pathogens

Probiotic microbes have long been acknowledged to possess antimicrobial properties against human and animal pathogens [1,48,49]. LabMAX-3, used in this study, contained three probiotic cultures of *Lactobacillus acidophilus*, *Lacticaseibacillus casei*, and *Enterococcus faecium* [39,40,41]. The identity of each culture was verified by PCR targeting the 16S rRNA gene (Figure 1) and by light microscopy (Appendix A) to be *Lactobacillus acidophilus*, *Lacticaseibacillus casei*, and *Enterococcus faecium*.

The antimicrobial activity of each live probiotic culture and the LabMAX-3 showed strong inhibitory zones against both Gram-negative *Salmonella enterica* serovar Typhimurium, enterotoxigenic *Escherichia coli* (ETEC) strains F4 and O78:H11, *Campylobacter jejuni* ATCC 29428, and Gram-positive *L. monocytogenes* (Figure 2, Table 3). The zone of inhibition produced by LabMAX-3 was comparable to zones produced by *L. acidophilus*, *E. faecium*, and *P. acidilactici*. *L. casei* showed the weakest antimicrobial response among the lactic acid bacterial cultures tested (Table 3). Surprisingly, neither heat-killed probiotic bacteria nor cell-free culture supernatants of the probiotic bacteria showed any growth inhibition of test pathogens except for heat-killed *L. acidophilus* cells, which showed a faint zone of inhibition against *Listeria*, *Salmonella*, and ETEC (Appendix A). On the other hand, cell-free supernatant from *P. acidilactici*, a known bacteriocin producer [46], inhibited *L. monocytogenes* and *S. aureus* but not *Salmonella* (Figure 2) since bacteriocin (Pediocin AcH) produced by *P. acidilactici* is effective only against Gram-positive bacteria. These results indicate that live probiotic bacteria produce inhibitory compounds, including acids, hydrogen peroxide, and others, that are effective against the Gram-positive and Gram-negative pathogens tested. Among the live cultures tested, *P. acidilactici* is a known bacteriocin producer and showed the highest inhibition of target pathogens, indicating possible synergistic effects of bacteriocin and other antimicrobials. Since heat treatment or cell-free culture supernatants of LabMAX-3 did not show an effect against both Gram-positive and Gram-negative pathogens, this suggests that interaction with live cultures is essential for suppressing pathogen growth (Appendix A).

Probiotic bacteria inhibit pathogens by competing with colonization sites, modulating microbial communities, and inhibiting pathogen growth by producing acids, hydrogen peroxide, bacteriocins, and other inhibitors [1,49]. Since cell-free culture supernatants from LabMAX-3 showed no inhibition compared to the *P. acidilactici* supernatant, bacteriocin may not significantly contribute to the inhibitory effects. Previous studies have demonstrated that probiotic supplementation can reduce hemorrhagic diarrhea and inflammatory bowel disease-like symptoms in dogs [2,50,51]. Panja et al. [52] reported that feeding a probiotic-supplemented diet to dogs increased creatinine levels in the serum. Likewise, a probiotic blend of *Lacticaseibacillus casei* Zhang, *Lactiplantibacillus plantarum* P-8, and *Bifidobacterium animalis* improved cytokine profiles and serum immunoglobulins and altered microbial community. The microbiota profile showed increased *Lactobacillus* spp. and *Faecalibacterium prausnitzii* presence and decreased *Sutterella stercoricanisn* and *Escherichia coli* in the feces [40]. *Lacticaseibacillus rhamnosus* GG (LGG) also reduced *Escherichia coli* (EHEC) O157:H7-mediated attachment-effacement lesion and improved tight junction integrity by redistributing claudin-1 and Zo-1 in the MDCK cell line [53].

### 3.2. LabMAX-3 Forms Biofilm on Abiotic Surface

The analysis of biofilm formation by probiotic blend LabMAX-3 on a polystyrene 96-well microtiter plate, as measured by crystal violet staining, showed a gradual increase in cell mass over 72 h (Figure 3). LabMAX-3-mediated biofilm formation was compared with *S. aureus* and *L. casei* ATCC334. At 48 h, biofilm produced by *S. aureus* was significantly higher than LabMAX-3 (*p* < 0.05), while at 72 h, there were no statistical differences among the three cultures. Further analysis shows LabMAX-3 and *L. casei* ATCC334 biofilm mass grew over 72 h and followed similar growth kinetics (Figure 3A,B).

Biofilm formation was also quantified by counting cells in the biofilm. LabMAX-3 bacterial counts gradually increased from 24 h to 72 h, showing about 1 log increase (Figure 3C,D).

Light microscopic analysis of crystal violet-stained biofilms on glass slides shows spatial bacterial arrangement within the biofilms (Figure 4). LabMAX-3 forms multilayered biofilms on glass surfaces with dense and sparse areas. Close examination of biofilm architecture in the sparse area (Figure 4, right panel) revealed the dominance of rod-shaped cells (lactobacilli) with a few embedded coccoid cells (*E. faecium*). Furthermore, the LabMAX-3-mediated biofilm architecture appeared very similar to that of *L. casei* ATCC334. In contrast, *S. aureus* formed multilayered dense biofilms, and only a few dispersed individual cells (cocci) are visible near the dense cell mass of the biofilm (right panel). These data indicate that a probiotic LabMAX-3 blend produces multilayered mixed-culture biofilms with dense and light areas comparable to biofilms produced by *L. casei* ATCC334 and *S. aureus*.

Probiotic biofilm formation is critical for ensuring colonization in the gut [54]. Probiotics in LabMAX-3 formed robust biofilms in 72 h on the abiotic surfaces, comparable to *L. casei* and *S. aureus.* These data indicate that *L. casei* in the LabMAX-3 mix would likely contribute to biofilm cell mass by producing glycocalyx [55]. Similarly, the contribution of *L. acidophilus* and *E. faecium* cannot be ruled out, akin to previous reports [56,57].

### 3.3. LabMAX-3 Probiotic Blend Adhesion and Biofilm Formation on MDCK Cells

We also examined the adhesion and biofilm-forming characteristics of LabMAX-3 in MDCK cell monolayers pretreated with or without LPS. LPS was used to simulate inflamed conditions [43]. LabMAX-3 adhesion to control MDCK cells (without LPS pretreatment) was calculated to be about 6.7 log CFU/mL, and similar counts (~6.8 log CFU/mL) to MDCK were observed when MDCK cells were pretreated with LPS (Figure 5). In contrast, *L casei* showed slightly higher adhesion (7.1 log CFU/mL) to MDCK cells with or without LPS treatment.

LDH release from MDCK cells during LabMAX-3 adhesion was analyzed, and values were below zero, indicating that LabMAX-3 treatment did not cause cytotoxicity (Figure 5B). Further microscopic analysis of cell monolayers did not reveal any visible cell damage, and the monolayer integrity remained intact during LabMAX-3 exposure (Figure 5C). These data indicate that the probiotic blend LabMAX-3 efficiently adheres to MDCK cells without causing any cell damage, and the adhesion is comparable to *L. casei* ATCC334 [14]. These data further suggest that LabMAX-3 can interact with healthy and LPS-treated inflamed tissue to promote canine gut health.

Next, we analyzed the adhesion patterns of LabMAX-3 to MDCK cells by Giemsa staining (Figure 5D). The data show that LabMAX-3 attaches to MDCK cells, forming patchy biofilm-like structures throughout the monolayer. *L. casei* ATCC334 also forms similar patchy biofilm-like structures on MDCK cells at a slightly higher frequency. The biofilms produced by LabMAX-3 and *L. casei* on MDCK follow a similar trend to the adhesion counts reported in Figure 5A. These data indicate that the adhesion of LabMAX-3 is facilitated by forming biofilms, ensuring prolonged persistence and potential health benefits to the host.

The ability of probiotic strains to produce biofilms on host epithelial cells is critical for ensuring colonization, pathogen exclusion, and promoting health [14,16,54,58]. In general, probiotic bacteria in the gut are transient, and their poor colonization in an inflamed gut environment may explain their inconsistent health benefits [59]. Therefore, LPS pretreatment on the MDCK cell line was used to create an inflamed condition in the gut [43]. LabMAX-3 could adhere and form biofilms on MDCK cell monolayers irrespective of LPS treatment.

### 3.4. Anti-Inflammatory Response of LabMAX-3 to MDCK Cells

We quantified the proinflammatory (TNFα) and anti-inflammatory (IL-10 and TGFβ) cytokine profiles of LabMAX-3 on MDCK monolayers (Figure 6). TNFα expression in LPS-pretreated MDCK cells exposed to LabMAX-3 was significantly higher (*p* = 0.005) than in cells without LPS treatment. A similar result was also observed for *L. casei* ATCC334. In contrast, a reduced IL-10 level (numerical difference but not a statistically significant difference) was noticed in MDCK cells pretreated with LPS compared to those without LPS treatment following LabMAX-3 exposure. A similar trend was observed with *L. casei* ATCC334 treatment. The IL-10 and TNFα ratio analysis demonstrated a significant (*p* < 0.05) reduction in values in LPS-pretreated MDCK cells compared to those with LPS treatment following LabMAX-3 exposure. TGFβ secretion showed no significant differences between LabMAX-3 and *L. casei* ATCC334 treatment. Cytokine profile data indicate LabMAX-3 has an anti-inflammatory response towards MDCK cells consistent with probiotic functionality reported by others [3,60].

## 4. Conclusions

Probiotic supplementation for systemic health in canines is a growing market. Here, we explored LabMAX-3 as a novel kibble additive comprising three renowned probiotics: *Lactobacillus acidophilus*, *Enterococcus faecium*, and *Lacticaseibacillus casei*. LabMAX-3 successfully inhibited the growth of notable canine-associated pathogens (*Salmonella enterica*, *Listeria monocytogenes*, and enterotoxigenic *Escherichia coli*). LabMAX-3 was shown to be a strong biofilm producer on both abiotic and biotic surfaces. The ability of LabMAX-3 to form biofilm-like structures and its strong adhesion to epithelial MDCK cell monolayers suggests its potential to improve canine gut health. LabMAX-3 showed no cytotoxic effects on the MDCK cell line with and without LPS stimulation. The anti-inflammatory properties of LabMAX-3 promote higher IL-10 and TNFα ratios when compared to the *L. casei* strain ATCC334 alone. The benefit of a multi-probiotic mixture as a dietary supplement is that it increases the chance of a positive effect on health in case one or more members fail to grow in an unfavorable host environment. Together, these data indicate that LabMAX-3 is viable as a supplement to companion animal kibble for promoting systemic health. The inclusion of other health-promoting probiotic strains in the probiotic blends may be considered for broadening probiotic function.

## Figures and Tables

**Figure 1 microorganisms-12-02284-f001:**
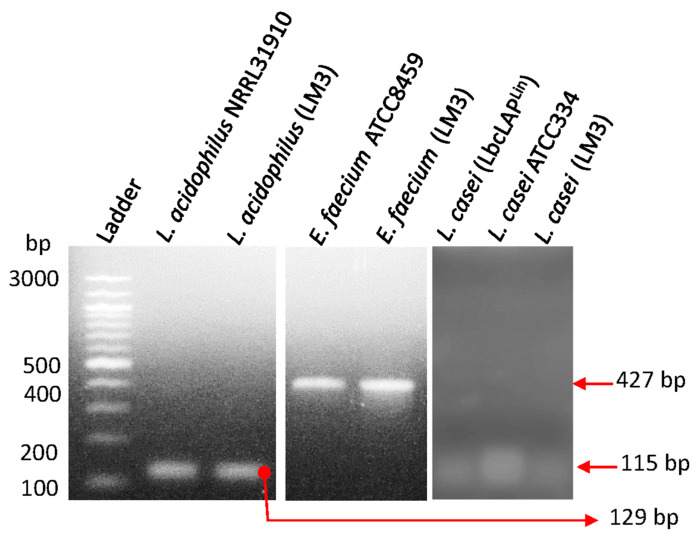
PCR confirmation of individual probiotic cultures from canine kibble supplement LabMAX-3 (LM3) (see Table 1). The remaining cultures were used as positive controls.

**Figure 2 microorganisms-12-02284-f002:**
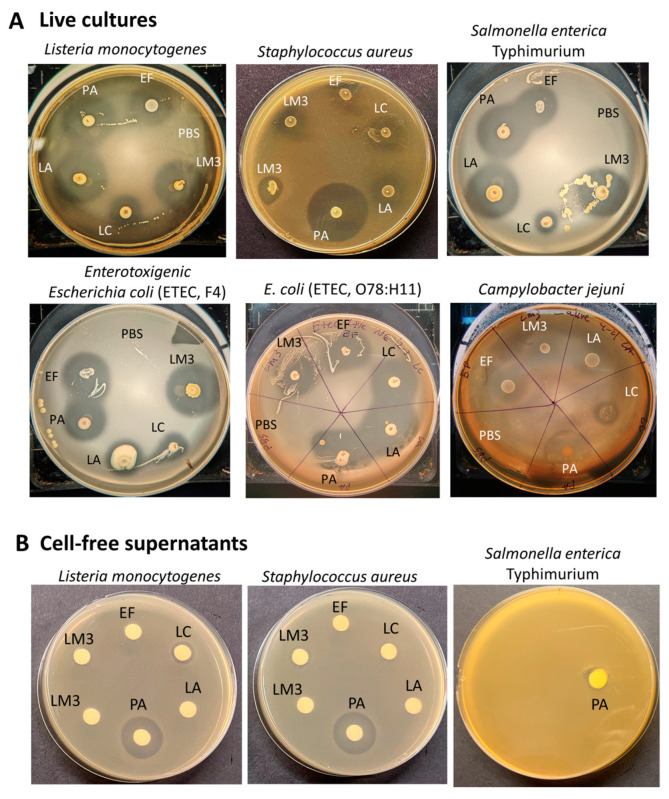
Antimicrobial activity of live LabMAX-3 and other lactic acid bacterial strains against pathogens. (**A**) Antimicrobial activity of 18 h MRS plate-grown cultures against pathogens. (**B**) Antimicrobial activity of cell-free culture supernatants of LabMAX-3 and other lactic acid bacterial strains against *L. monocytogenes*, *S. aureus*, and *Salmonella*. Antimicrobial activity was observed after 24 h. LM3, LabMAX-3; LA, *Lactobacillus acidophilus*; LC, *Lacticasiebacillus casei*; EF, *Enterococcus faecium*; PA, *Pediococcus acidilactici*.

**Figure 3 microorganisms-12-02284-f003:**
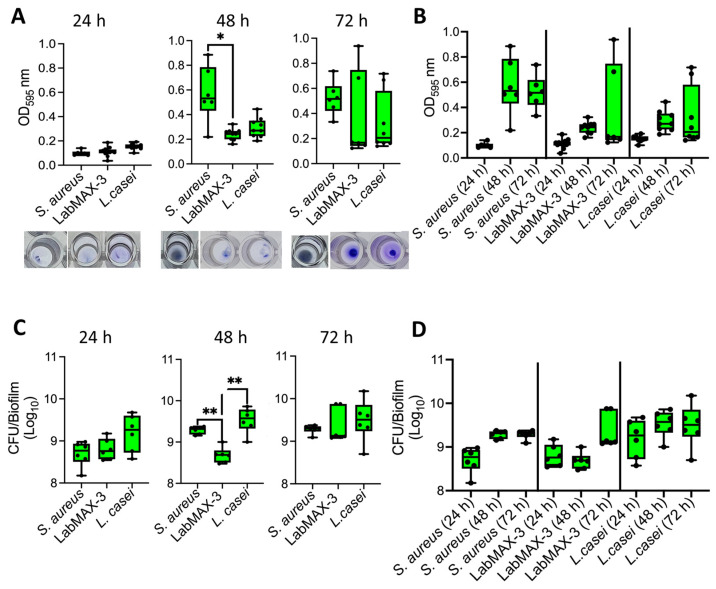
Biofilm formation by probiotic blend LabMAX-3 quantified by (**A**,**B**) crystal violet staining and (**C**,**D**) counting colony-forming units (CFU) after 24, 48, and 72 h incubation at 37 °C. Biofilm formation by LabMAX-3 was compared with *Staphylococcus aureus* and *Lacticaseibacillus casei* ATCC334. *, *p* < 0.05; **, *p* < 0.005.

**Figure 4 microorganisms-12-02284-f004:**
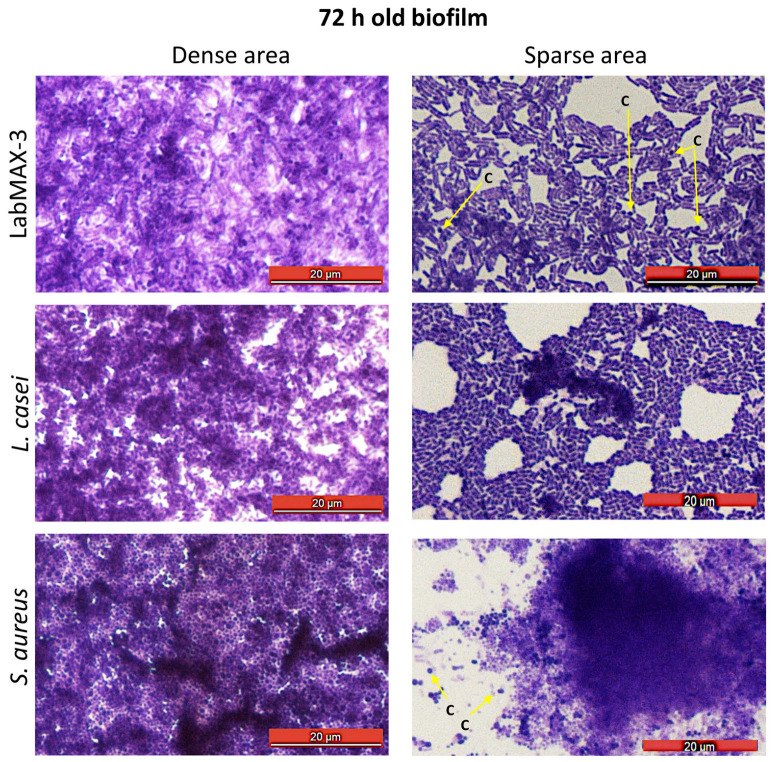
Microscopic examination of 72 h old biofilms from dense and sparse areas of biofilms after staining with crystal violet. Arrows pointing to coccoid cells (c).

**Figure 5 microorganisms-12-02284-f005:**
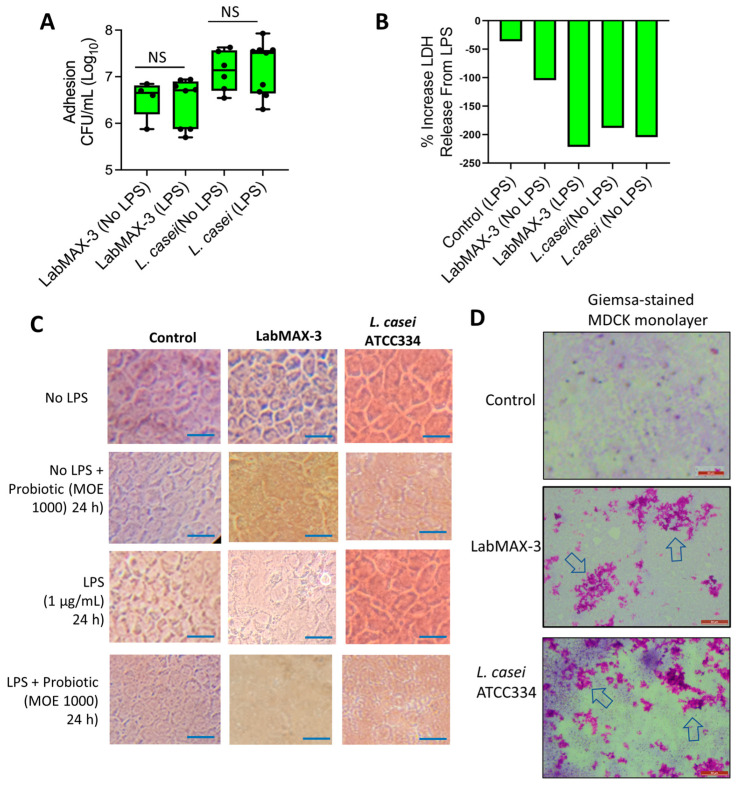
Adhesion characteristics of LabMAX-3 in the MDCK cell line. (**A**) Adhesion (CFU/mL) of probiotic cultures to MDCK cells. (**B**) Lactate dehydrogenase (LDH) release assay from MDCK cells during the LabMAX-3 adhesion experiment. (**C**) Light microscopic analysis of cell monolayer integrity during probiotic adhesion experiment. Scale: 25 µm. (**D**) Giemsa staining of MDCK cells after probiotic exposure. LabMAX-3 forms biofilm-like structures in patches (arrows) on MDCK cell monolayers. Scale: 50 µm.; NS, not significant.

**Figure 6 microorganisms-12-02284-f006:**
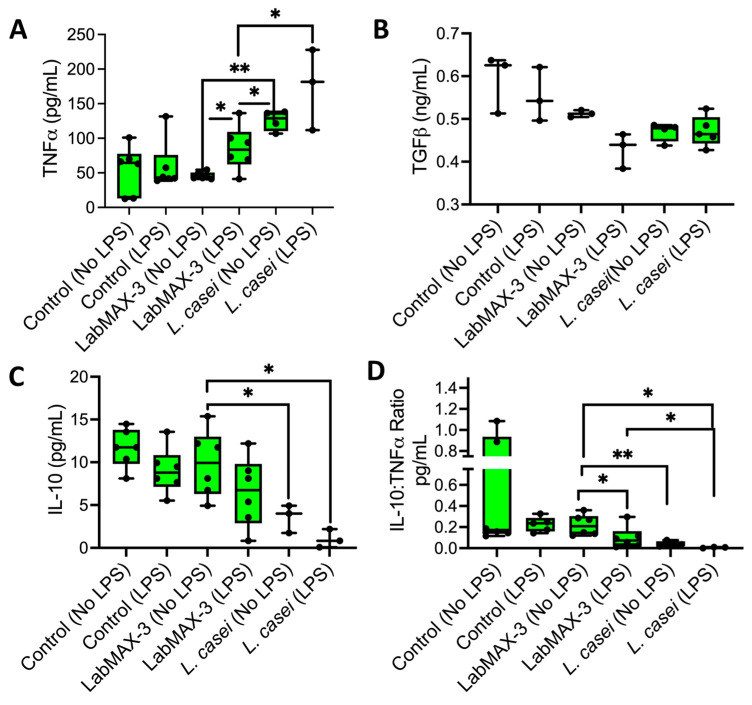
Cytokine secretion from the MDCK cell line by probiotics in the presence or absence of LPS pretreatment: (**A**) TNFα, (**B**) TGFβ, (**C**) IL-10, (**D**) IL-10:TNFα ratio. Data are an average of 2 experiments in triplicate. *, *p* < 0.05; **, *p* < 0.005.

**Table 1 microorganisms-12-02284-t001:** PCR confirmation of probiotic strains.

Organism	Primer	Target Gene	Product Size (bp)	Ref.
*L. acidophilus*	F-CCT TTC TAA GGA AGC GAA GGA T R-ACG CTT GGT ATT CCA AAT CGC	16S–23S	129	[44]
*L. casei*	F-CCA CAA TCC TTG GCT GTT CT R-GCT TGA GGC GAT TGT AAT CC	Putative protein	115	[44]
*E. faecium*	F-GCGTGCATGGTTAAGACGAC R-CTGCTGGATCGCTGGGTTAT	Rhomboid protease GluP (serine protease)	427	[45]

**Table 2 microorganisms-12-02284-t002:** Bacterial cultures used in the study.

Bacteria	Identification	Source	Provided By
Enterotoxigenic *Escherichia coli*	F4 (K88)	Swine rectal isolate	Paul Ebner/Animal Sciences Purdue University
Enterotoxigenic *Escherichia coli*	O78:H11	Human fecal isolate	ATCC 35401
*Listeria monocytogenes*	F4244	Clinical isolate, Human Central Nervous System (CNS) isolate	CDC, Our lab collection
*Campylobacter jejuni*	ATCC 29428	Human fecal isolate	ATCC
*Salmonella enterica* serovar Typhimurium	ST-1		Our collection
*Lactobacillus acidophilus*	LabMAX-3	CH2 Animal Solutions	CH2 Animal Solutions
*Lacticaseibacillus casei*	LabMAX-3	CH2 Animal Solutions	CH2 Animal Solutions
*Enterococcus faecium*	LabMAX-3	CH2 Animal Solutions	CH2 Animal Solutions
*Lactobacillus acidophilus*	NRRL 31910	ATCC	Our lab collection
*Lacticaseibacillus casei*	ATCC 334	Cheese	Our collection
*Enterococcus faecium*	ATCC 8459	Cheese	Dharmendra Mishra, Purdue University
*Pediococcus acidilactici*	H	Fermented sausage	Our lab collection [46]
*Staphylococcus aureus*	ATCC 25923 (Rosenbach)	Clinical isolate	Our lab collection

**Table 3 microorganisms-12-02284-t003:** LabMAX-3-mediated inhibition of pathogens.

Bacteria	Strain	Avg Zone of Inhibition ± SEM (mm)					
		LabMAX-3	LC *	LA *	EF *	PA	LC (ATCC 334)
*Listeria monocytogenes*	F4244	15.83 ± 1.64	8.50 ± 0.00	19.00 ± 1.44	17.67 ± 0.72	20.67 ± 0.44	11.00 ± 0.57
*Staphylococcus aureus*	ATCC 25923	7.33 ± 0.67	10.67 ± 0.67	9.33 ± 0.67	7.33 ± 0.67	20.67 ± 0.67	18.33 ± 0.88
*Salmonella enterica* serovar Typhimurium	ST-1	22.67 ± 1.59	10.33 ± 0.33	15.50 ± 0.50	18.00 ± 1.26	19.83 ± 0.33	9.33 ± 2.19
Enterotoxigenic *Escherichia coli*	F4 (K88)	19.83 ± 1.69	8.17 ± 0.88	21.17 ± 0.17	19.50 ± 1.041	22.67 ± 0.17	13.67 ± 2.73
Enterotoxigenic *Escherichia coli*	O78:H11	6.00 ± 0.57	9.33 ± 0.67	10.00 ± 1.00	9.00 ± 0.00	11.67 ± 0.33	11.67 ± 3.28
*Campylobacter jejuni* ⁋	ATCC 29428	15.00	10.00	10.00	13.50	22.00	NA

LabMAX-3 contains EF, LA, and LC (1:1:1); * denotes origin from LabMAX-3; EF, *Enterococcus faecium*; LA, *Lactobacillus acidophilus*; LC, *Lacticaseibacillus casei*; PA, *Pediococcus acidilactici*; and LC (ATCC 334), *Lacticaseibacillus casei* ATCC 334. ⁋ Indicates one trial.

## Data Availability

All data are presented in the manuscript.

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
