# Peer review of "Assessment of Biofilm Formation and Anti-Inflammatory Response of a Probiotic Blend in a Cultured Canine Cell Model"

_microorganisms, 2024, doi:10.3390/microorganisms12112284_

Round 1
Reviewer 1 Report
Comments and Suggestions for Authors
Dear authors
Thanks for your work and presentation.
Please, refer to the following comments during your manuscript revision.
- In title, the terms “in vitro” and “used in dogs” could be added to the title.
- Abbreviations should be mentioned as full words at the first mention, and then mentioned as abbreviations. This should be followed in the abstract and in the whole manuscript.
- Some another keywords should be supplied rather than those in the title.
- In the introduction, the section contains lines 59-64, should be added after lines 65-72.
- The aim of the study should be the last lines in the introduction.
- In materials and methods section, a lot of data were mentioned in details. If the methods are repeated, references could be enough.
- Some information are repeated in the conclusion section. It should be concise and supplemented with some recommendations.
Best wishes
Author Response
Review #1
1. In title, the terms “in vitro” and “used in dogs” could be added to the title.
We have revised the title to reflect the study objectives - “Assessment of Biofilm Formation and Anti-Inflammatory Response of a Probiotic Blend in a Cultured Canine-Cell Model.”
2. Abbreviations should be mentioned as full words at the first mention, and then mentioned as abbreviations. This should be followed in the abstract and in the whole manuscript.
We have carefully checked and revised all abbreviations as suggested throughout the manuscript.
3. Some another keywords should be supplied rather than those in the title.
We have added additional keywords: MDCK; Cultured cell model; Lactobacillus; Enterococcus
4. In the introduction, the section contains lines 59-64, should be added after lines 65-72. The aim of the study should be the last lines in the introduction.
We have thoroughly revised, shortened and reorganized the introduction section for improved clarity.
5. In materials and methods section, a lot of data were mentioned in details. If the methods are repeated, references could be enough.
We reviewed the materials and methods section closely; however, we did not find the concern the reviewer raised. Because of page limitations in some journals, authors are often forced to shorten the description of the methods, which sometimes hinders others from reproducing the experiments accurately. Since this journal (Microorganisms) does not have such restrictions, we decided to provide the details with a reference where appropriate. We hope this is acceptable to the reviewer.
6. Some information are repeated in the conclusion section. It should be concise and supplemented with some recommendations.
We have revised the conclusion section, provided a recommendation statement and removed redundant information (Lines 372-387).
Reviewer 2 Report
Comments and Suggestions for Authors
1. Could you please revise the title; it looks title of a book
2.Ln 103 -105; please revise or move or add a reference
3. Ln 236, in my opinion it is better to move this sentence to the section of material and methods
4. Please revise the conclusion to be direct and clear in 2 to 3 sentences.
Author Response
1. Could you please revise the title; it looks title of a book
We have revised the title to accurately reflect the study objectives - “Assessment of Biofilm Formation and Anti-Inflammatory Response of a Probiotic Blend in a Cultured Canine-Cell Model.”
2. Ln 103 -105; please revise or move or add a reference
As suggested, the Introduction section has been thoroughly revised with additional new references (see references 39-41).
3. Ln 236, in my opinion it is better to move this sentence to the section of material and methods
As suggested, we have moved this sentence to the material and methods section.
4. Please revise the conclusion to be direct and clear in 2 to 3 sentences.
We have revised the conclusion section and removed some statements to avoid redundancy.
Reviewer 3 Report
Comments and Suggestions for Authors
The present study has revealed a commercial probiotic blend (LabMAX-3) exerts anti-inflammatory effect on MDCK cell line. The results have suggested that LabMAX-3 can be applied to improve gut health in dogs. The data could provide important information for pet sciences. However, major revisions are required for publication, following are some comments on the manuscript.
Introduction: The introduction part is a bit long. The authors are suggested to reduce the length of introduction. The introduction of probiotic in the first and second paragraphs could be shortened. The fourth paragraph could also be combined with the first and second paragraphs to introduce the probiotic.
Line 73-77: Please add text about the effects of certain probiotic (Enterococcus faecium, Lactobacillus acidophilus, or Lacticaseibacillus casei) involved in LabMAX-3 on the gut health of canine.
Line 108-110: Please add the exact content of each probiotic in commercial LabMAX-3. Which is the main component? The text in Line 237-243 should be put in Materials and Methods.
Table 3: Why LA did not show any inhibitory effect on Campylobacter jejuni? Authors are suggested to discuss it in the text.
Figure 2: Why the left part of Salmonella culture dish is not shown in Fig 2B?
Line 302: Fig 3 is missing in the manuscript.
Figure 4: What does the blue arrow represent? It should be specified in caption of Fig 4.
Figure 5: The size of pictures in Fig 5C are different. Scale bars are missing. The resolution of Fig 5C and D is low. The pictures of LPS group seems to be duplicated. Please revise it.
Line 340-341: No visible cell damage could be observed in the control between the No LPS and LPS groups. The MDCK cells should be damaged by LPS treatment.
Line 374: Fig 6 is also missing, which makes it difficult to read the manuscript.
Comments on the Quality of English LanguageModerate editing of English language required.
Author Response
The present study has revealed a commercial probiotic blend (LabMAX-3) exerts anti-inflammatory effect on MDCK cell line. The results have suggested that LabMAX-3 can be applied to improve gut health in dogs. The data could provide important information for pet sciences. However, major revisions are required for publication, following are some comments on the manuscript.
1. Introduction: The introduction part is a bit long. The authors are suggested to reduce the length of introduction. The introduction of probiotic in the first and second paragraphs could be shortened. The fourth paragraph could also be combined with the first and second paragraphs to introduce the probiotic.
We have significantly shortened, reorganized and revised the Introduction section.
2. Line 73-77: Please add text about the effects of certain probiotic (Enterococcus faecium, Lactobacillus acidophilus, or Lacticaseibacillus casei) involved in LabMAX-3 on the gut health of canine.
We have added supporting references to reflect their contribution to canine gut health (see Refs 39-41).
3. Line 108-110: Please add the exact content of each probiotic in commercial LabMAX-3. Which is the main component? The text in Line 237-243 should be put in Materials and Methods.
The LabMax-3 is a proprietary formulation containing live probiotics and other components that the manufacturer is unwilling to disclose in detail. However, they indicated that the product contains equal proportions of three probiotics; hence, we tested the blends with all three probiotics in equal concentrations.
The text in lines 237-243 is now placed in the materials and methods section (see lines 102-105).
4. Table 3: Why LA did not show any inhibitory effect on Campylobacter jejuni? Authors are suggested to discuss it in the text.
This was a typo; it should be 10 cm and not 0 cm (corrected). Since there was inhibition, which is self-explanatory and does not require further discussion.
5. Figure 2: Why the left part of Salmonella culture dish is not shown in Fig 2B?
The other half of the plate was blank. Nothing was tested in that part, which is why it was cropped. Now, the complete picture of the plate is provided (see Figure 2)
6. Line 302: Fig 3 is missing in the manuscript.
We are also surprised that Fig 3 was mysteriously omitted from the version of the manuscript the reviewers received. The original submission had all the figures posted in the preprint (https://www.preprints.org/manuscript/202409.0922). Please see Figure 3 in the revised version.
7. Figure 4: What does the blue arrow represent? It should be specified in caption of Fig 4.
As stated in the Figure caption, arrows pointed to coccoid cells. Color arrows had no significance. Different colors were used for improved visibility against the background. Now, uniform color is used (see revised Fig 4).
8. Figure 5: The size of pictures in Fig 5C are different. Scale bars are missing. The resolution of Fig 5C and D is low. The pictures of LPS group seems to be duplicated. Please revise it.
We have reconfigured the figure panels and added scale bars. Unfortunately, we do not have high-resolution images for panels C and D. With the imposing time constraint, generating these figures won't be easy since the cell line takes several days to grow, and reseeding in slide-flaskets would take several more. We agree that higher resolution would improve the figure quality, but we do not believe the data would differ.
Thank you for pointing out the duplication of panels in the LPS treatment group. It was our oversight. We have replaced the panel in question with the correct image (See Fig 5C).
9. Line 340-341: No visible cell damage could be observed in the control between the No LPS and LPS groups. The MDCK cells should be damaged by LPS treatment.
The LPS was used at 1 µg/mL, and this concentration is too low to induce cell damage but is sufficient to induce inflammation (Capilliano et al. 2020).
10. Line 374: Fig 6 is also missing, which makes it difficult to read the manuscript.
The original manuscript we submitted had all the figures, but it is unfortunate to learn that the version the reviewers received did not have Fig 3 and 6. Please see the missing figures in the revised version of the manuscript.
Reviewer 4 Report
Comments and Suggestions for Authors
Line 173: the specific iSonic sonicator model and waveform setting should be listed.
Section 2.4: the authors reported mixing the individual probiotic cultures at a 1:1:1 ratio, it is unclear whether the three probiotic cultures were also mixed at the same ratio in LabMAX. The CFU/g of probiotics in feed and their ratio might influence the study outcomes, e.g., those reported in Figure 4.
Lines 248 to 264: the authors indicated that the antimicrobial activity of each live probiotic culture showed strong inhibitory zones against test pathogens while cell-free supernatant exhibited limited activities. As evident from Figures 2A and 1B, PA exhibited the highest antimicrobial properties against L. monocytogenes, S. aureus, and S. enterica Typhimurium. While the cell-free supernatant only showed limited activities from PA, it suggests that the concentration of antimicrobial(s) in the cell-free supernatant is lower than those in the live cultures. If the authors concentrate the cell-free supernatants and conduct the experiments again, it might paint a different picture. Furthermore, the authors suggested on Line 263 that interaction with live cultures is essential for suppressing pathogens’ growth. As evident from Figure 2A, the authors should discuss the synergetic interactions among the antimicrobials produced by the probiotic strains.
Figure 5C: the authors should include scale bars for individual images.
Author Response
1. Line 173: the specific iSonic sonicator model and waveform setting should be listed.
The specifics of iSonic info (Model #P4830, set at Frequency 60 hz, Watt 150, Volt 110-120, waveform 18.3; Chicago, IL, USA) have been provided (see Line 168).
2. Section 2.4: the authors reported mixing the individual probiotic cultures at a 1:1:1 ratio, it is unclear whether the three probiotic cultures were also mixed at the same ratio in LabMAX. The CFU/g of probiotics in feed and their ratio might influence the study outcomes, e.g., those reported in Figure 4.
Because the LabMAX-3 is a proprietary blend, without disclosing detailed formulation, they indicated the product contained equal ratios of three probiotics thus we used 1:1:1. We certainly agree with the reviewer that if probiotics are used at different ratios, they may influence the response. This has been explicitly stated in the revised manuscript (See lines 100-105).
3. Lines 248 to 264: the authors indicated that the antimicrobial activity of each live probiotic culture showed strong inhibitory zones against test pathogens while cell-free supernatant exhibited limited activities. As evident from Figures 2A and 1B, PA exhibited the highest antimicrobial properties against L. monocytogenes, S. aureus, and S. enterica Typhimurium. While the cell-free supernatant only showed limited activities from PA, it suggests that the concentration of antimicrobial(s) in the cell-free supernatant is lower than those in the live cultures. If the authors concentrate the cell-free supernatants and conduct the experiments again, it might paint a different picture. Furthermore, the authors suggested on Line 263 that interaction with live cultures is essential for suppressing pathogens’ growth. As evident from Figure 2A, the authors should discuss the synergetic interactions among the antimicrobials produced by the probiotic strains.
We agree with the reviewer that we may observe a different result if the supernatants were concentrated. We deliberately avoided doing so because that would not reflect the natural situation during probiotic feeding. We also agree that the higher antimicrobial activity of live cultures reflects the potential synergistic activity of antimicrobials produced during growth. We have included a statement in the revised document (Line 251-256).
4. Figure 5C: the authors should include scale bars for individual images.
We have included a scale bar in Figure 5C.
Round 2
Reviewer 2 Report
Comments and Suggestions for Authors
The authors adopted all comments
Reviewer 3 Report
Comments and Suggestions for Authors
The manuscript has been improved.